# Regenerating Myofibers after an Acute Muscle Injury: What Do We Really Know about Them?

**DOI:** 10.3390/ijms241612545

**Published:** 2023-08-08

**Authors:** Francis X. Pizza, Kole H. Buckley

**Affiliations:** 1Department of Exercise and Rehabilitation Sciences, University of Toledo, Toledo, OH 43606, USA; 2Division of Gastroenterology and Hepatology, University of Pennsylvania Perelman School of Medicine, Philadelphia, PA 19104, USA; kole.buckley@pennmedicine.upenn.edu

**Keywords:** muscle regeneration, muscle repair, embryonic myogenesis, postnatal myogenesis

## Abstract

Injury to skeletal muscle through trauma, physical activity, or disease initiates a process called muscle regeneration. When injured myofibers undergo necrosis, muscle regeneration gives rise to myofibers that have myonuclei in a central position, which contrasts the normal, peripheral position of myonuclei. Myofibers with central myonuclei are called regenerating myofibers and are the hallmark feature of muscle regeneration. An important and underappreciated aspect of muscle regeneration is the maturation of regenerating myofibers into a normal sized myofiber with peripheral myonuclei. Strikingly, very little is known about processes that govern regenerating myofiber maturation after muscle injury. As knowledge of myofiber formation and maturation during embryonic, fetal, and postnatal development has served as a foundation for understanding muscle regeneration, this narrative review discusses similarities and differences in myofiber maturation during muscle development and regeneration. Specifically, we compare and contrast myonuclear positioning, myonuclear accretion, myofiber hypertrophy, and myofiber morphology during muscle development and regeneration. We also discuss regenerating myofibers in the context of different types of myofiber necrosis (complete and segmental) after muscle trauma and injurious contractions. The overall goal of the review is to provide a framework for identifying cellular and molecular processes of myofiber maturation that are unique to muscle regeneration.

## 1. Introduction

Trauma, strenuous physical activity, or disease can disrupt the structure of myofibers, the principal cell type within skeletal muscle. Structural abnormalities and the associated muscle dysfunction, soreness, and edema serve as signs and symptoms of an injury [1,2,3,4,5]. Hallmark signs of myofiber injury include lesions in the cytoskeleton and increased sarcolemma permeability [2,3,6,7,8,9,10,11,12,13,14]. Myofiber injury can vary in severity, ranging from minor to extensive disruptions to the cytoskeleton and sarcolemma. When severely injured, the entire myofiber or just a segment of it undergoes necrosis. The number of affected myofibers and the extent to which they display signs of injury or necrosis are predictors of muscle dysfunction after injury [3,11,14,15].

Myofiber injury, regardless of the severity, initiates processes that are intended to either restore structure to injured myofibers (myofiber repair) or generate new myofibers to replace necrotic myofibers (muscle regeneration) [16,17,18,19,20]. Timely completion of myofiber repair and muscle regeneration is essential to restoring structure and function to injured muscle as delays or impairments can result in prolonged deficits in strength and motility. Such deficits can impair performance of daily activities, predispose individuals to a sedentary lifestyle, and exacerbate preexisting conditions. There is a long standing interest in expanding knowledge of processes that restore structure and function to injured muscle in the hopes of preventing deleterious outcomes stemming from a muscle injury.

The majority of injury resolution research has focused on detailing cellular and molecular processes of the early stages of muscle regeneration. These stages are critically important to restoring muscle structure as they give rise to new, immature myofibers called regenerating myofibers. Regenerating myofibers differ from normal myofibers in that they contain nuclei (myonuclei) in a central, as opposed to a peripheral, location near the sarcolemma. During the later stages of regeneration, regenerating myofibers are thought to mature into normal sized myofibers with peripheral myonuclei. Arguably, regenerating myofiber maturation is as important as their formation in restoring structure to injured muscle. Despite an extensive body of literature on muscle regeneration, fundamental processes that govern the maturation of regenerating myofibers are not well understood.

The primary objective of this narrative review is to highlight the phenotype of regenerating myofibers after an acute muscle injury and to summarize evidence pertaining to their maturation. We begin with the semantics of commonly used terms and a perspective for how the proper use of nomenclature can foster a deeper understanding of muscle regeneration. Next, we present an overview of the muscle regeneration paradigm, which provides context for the bulk of the review. Introductory topics are followed by a discussion of similarities and differences in myofiber maturation during muscle development (embryonic and postnatal myogenesis) and regeneration. This is intended to address the widely held belief that processes facilitating myofiber maturation during embryonic and postnatal myogenesis are recapitulated during muscle regeneration. Lastly, we discuss regenerating myofibers in the context of complete and segmental necrosis of myofibers, different types of muscle trauma, as well as after injurious contractions. We intentionally minimize discussion of specific molecules that regulate aspects of muscle regeneration for two reasons. One, the regulation of the initial stages of muscle regeneration by molecules, cell signaling pathways, and cell types has been expertly summarized in several recent reviews [21,22,23,24,25,26,27]. Two, little is known about regenerating myofiber maturation after an acute muscle injury. The hope is that a focused review on the phenotype and physiology of regenerating myofibers, as well as perspectives on their maturation, will facilitate the discovery of cellular and molecular processes of regenerating myofiber maturation. 

## 2. Semantics of Nomenclature

The terms ‘muscle repair’ and ‘muscle regeneration’ are often used interchangeably to describe processes that restore structure and function to acutely injured skeletal muscle. They have also been used to describe processes that attempt to resolve abnormalities in atrophic, denervated, aged, or diseased muscle. This broad usage becomes problematic when trying to decipher overlapping and unique processes that regulate structure and function in acutely injured muscle and in chronic conditions affecting skeletal muscle. Specifically, it implies that processes that restore structure and function to acutely injured muscle are indistinguishable from those operating in atrophic, denervated, aged, or diseased muscle. Although similarities may exist, cellular and molecular processes intended to alleviate acute or chronic problems in skeletal muscle are, in general, targeted to the root of the problem. The nomenclature used should ideally encapsulate the problem, which provides context into processes that resolve it. Given the array of potential problems in skeletal muscle, this is easier said than done. 

An acute muscle injury is characterized by structural abnormalities in myofibers (e.g., increased sarcolemma permeability and focal lesions in the cytoskeleton), signs of myofiber necrosis (e.g., ruptured sarcolemma and extensive lesions in the cytoskeleton), and a sustained deficit in muscle function [1,2,3,4,5,6,7,8,9,10,11,12,13,14]. Other aspects of skeletal muscle (e.g., myotendinous junction, connective tissue, and blood vessels) may also exhibit structural and functional abnormalities in injured muscle. Each abnormality will need to be resolved if structure and function are to be restored. The abnormality under study and the associated processes that resolve it should be considered when selecting nomenclature. 

Ideally, ‘muscle repair’ or ‘injury resolution’ should be used when broadly describing processes that restore structure and function to acutely injured muscle. This includes restoring structure to injured myofibers, creating replacement myofibers, and/or restoring structure to other components of skeletal muscle (Figure 1). On the other hand, ‘muscle regeneration’ should be used to define processes that govern the formation and subsequent maturation of myofibers in injured skeletal muscle [17]. This definition of ‘muscle regeneration’ is similar to that of ‘myogenesis’, which also describes de novo myofiber formation. The difference between the terms is that ‘muscle regeneration’ is restricted to injured muscle; whereas ‘myogenesis’ is usually preceded by an adjective that provides context for where or when myofibers are forming and/or maturing. For example, ‘embryonic myogenesis’ is commonly used to describe myofiber formation and maturation in the embryo and fetus. ‘Postnatal myogenesis’ is used to describe myofiber maturation after birth to adulthood, as the number of myofibers is believed to be fixed around the time of birth [28,29,30,31]. ‘In vitro myogenesis’ is used to describe the early stages of myofiber formation in cultured cells (e.g., myoblasts). Other derivations of ‘myogenesis’ are ‘regenerative myogenesis’ and ‘adult myogenesis’, which in essence are analogous to ‘muscle regeneration’. 

It is important to consider the context in which myofibers are forming and maturing as intrinsic properties of myogenic cells are regulated by many molecules (e.g., cytokines) and cells (e.g., motor neurons, macrophages, and fibroblasts) within skeletal muscle [23,26,32,33,34]. As the environment of embryonic/fetal, postnatal, and injured skeletal muscle are distinctly different, it stands to reason that there are processes of myofiber formation and maturation that are unique to muscle regeneration. Appropriate use of ‘muscle regeneration’ and derivations of ‘myogenesis’ should reduce the extrapolation of findings from embryonic, fetal, postnatal, and/or in vitro studies to the events that follow an acute muscle injury. Such an extrapolation can result in a misguided understanding of muscle regeneration, as well as negate the pursuit of cellular and molecular processes that are unique to muscle regeneration. 

‘Myofiber repair’ defines processes that restore structure to injured myofibers [19,20,35]. Stated differently, the processes of myofiber repair prevent myofiber necrosis in injured muscle. The distinction between ‘myofiber repair’ and ‘muscle regeneration’ highlights the concept that some injured myofibers survive, whereas others do not. Myofibers, like other cells in the body, have an intrinsic ability to prevent their demise when injured [19,20,35]. However, this ability can be overwhelmed, resulting in myofiber necrosis. Thus, structural abnormalities in myofibers and the loss of myofibers are two distinct problems in injured muscle that need to be resolved if structure and function are to be restored. Consistent use of nomenclature should aid the identification of common (e.g., satellite cell proliferation) and unique processes associated with ‘muscle regeneration’ and ‘myofiber repair’ which in turn, will expand knowledge of cellular and molecular processes that restore structure and function to injured muscle. 

## 3. General Overview of Muscle Regeneration

Muscle regeneration is limited to muscle injuries that result in necrosis of one or more myofibers [16,17]. As discussed later in this review, signs of necrosis can be found throughout a myofiber (complete necrosis) or localized to one or more regions of a myofiber (segmental or focal necrosis) (see “Section 5”). Furthermore, the number of necrotic myofibers in injured muscle can vary widely, depending on the cause of the injury. Conceptually, the magnitude of muscle regeneration is proportional to the number of necrotic myofibers within the injured muscle. 

The classic paradigm for muscle regeneration in response to complete necrosis of myofibers can be partitioned into sequential stages (Figure 2). Stages 1 and 2 give rise to new, immature myofibers called regenerating myofibers, which are defined by the central position of myonuclei [16,17]. During the later stages of regeneration (stages 3–4), regenerating myofibers are thought to mature into myofibers that are morphologically and functionally indistinguishable from uninjured and non-regenerating (normal) myofibers. Myofibers formed as a result of muscle regeneration, in essence, replace necrotic myofibers in the injured muscle. 

This overview focuses on the later stages of muscle regeneration, which is intended to provide a conceptual framework for the bulk of this review. The evidence presented herein is derived primarily from studies that used chemical (e.g., intramuscular injection of cardiotoxin or barium chloride) trauma to induce muscle regeneration in rodents, which is the most popular experimental approach to study processes of muscle regeneration. Literature pertaining to in vitro, embryonic, and postnatal myogenesis is included in the overview to fill knowledge voids in the paradigm of muscle regeneration. 

### 3.1. Stage 1: Myogenic Cell Proliferation

The initial phase of muscle regeneration is characterized by the proliferation of muscle stem cells called satellite cells and their descendants (myoblasts) [16,18]. Molecular processes associated with myogenic cell proliferation during regeneration has been expertly summarized in recent reviews [21,27]. Satellite cell/myoblast proliferation is highest 3–7 days post-injury and is substantially reduced during prolonged recovery from chemical trauma [36,37,38]. Satellite cell/myoblast proliferation occurs within the basal lamina that surrounded necrotic myofibers, as well as in the interstitium after injury [37,38,39,40,41]. Satellite cell/myoblast proliferation can also occur in the absence of overt signs of myofiber injury or necrosis [42] and, hence, the need for muscle regeneration. In other words, myogenic cell proliferation does not always lead to regenerating myofiber formation.

The essential role of satellite cells/myoblasts proliferation in muscle regeneration can be viewed from two perspectives. One, they are the principal source of myonuclei in myofibers, including regenerating myofibers [37,43,44,45]. Two, satellite cells are required for the formation of regenerating myofibers after injury. Genetic models that substantially reduced satellite cells (Pax7+ cells) from skeletal muscle prior to injury demonstrated this requirement. Few, if any, regenerating myofibers were observed in injured muscles of mice nearly depleted of satellite cells [38,44,46,47,48]. 

### 3.2. Stage 2: Myoblast Differentiation, Adhesion, and Fusion

The second stage of muscle regeneration is characterized by the differentiation of myoblasts [21,49,50] and their adhesion [25,51,52] and fusion to each other [24,53,54,55]. Myoblast–myoblast fusion creates a skeletal muscle cell type called a myotube, which initially has two myonuclei. Nascent myotubes are observable during in vitro myogenesis but are difficult to identify or isolate in injured muscle because of their small size and fragility, as well as the complexity of the cellular milieu of injured muscle. 

The necessity of myoblasts to fuse to each other to form myotubes has been substantiated and extended through the genetic manipulation of genes that are uniquely expressed by myoblasts. Specifically, deletion of *myomaker* or *myomixer* (also called *myomerger* and *minion*) prevented myotube formation during in vitro and embryonic myogenesis [56,57,58,59]. Myomaker and myomixer-mediated myoblast–myoblast fusion is also required for muscle regeneration. That is, few, if any, regenerating myofibers were observed in injured muscles of mice conditionally depleted of *myomaker* or *myomixer* [56,60]. These seminal findings demonstrate that the fusogenic properties of myomaker and myomixer are required for nascent myotube formation during in vitro myogenesis, embryonic myogenesis, and muscle regeneration [24,56,57,58,59]. 

### 3.3. Stage 3: Maturation of Myotubes into Regenerating Myofibers

The third stage of muscle regeneration involves the maturation of nascent myotubes into regenerating myofibers. This stage is understudied because it is difficult to study nascent myotubes in injured muscle, and there is ambiguity in defining the point in which myotubes become regenerating myofibers. Nevertheless, processes that regulate myotube maturation during in vitro and embryonic myogenesis are thought to be operating after an acute muscle injury. 

Nascent myotubes in vitro and in the embryo/fetus add nuclei via myoblast fusion [61,62,63,64,65]. Myotubes during in vitro and embryonic myogenesis can also add myonuclei via myotube–myotube fusion [66,67,68,69,70]. The process through which myotubes or myofibers add myonuclei is called myonuclear accretion. Myonuclear accretion during in vitro and embryonic myogenesis is accompanied by the formation and maturation of myofibrils (myofibrillogenesis) and organelles, the assembly of a basal lamina, as well as a progressive increase in myotube length and diameter [61,62,66,69,70,71,72,73,74,75,76,77,78]. These are hallmark signs of myotube maturation. Myotube maturation may also be indicated by the repositioning of myonuclei. Recently added myonuclei normally cluster together during in vitro myogenesis and then disperse as myotubes increase in size [79,80,81]. The dispersion of myonuclei within the confines of a relatively small cytoplasmic area results in myonuclei appearing to be in a central position in developing myotubes [61,69,77,82]. 

At some point in their maturation, myotubes formed in vivo after injury [40,83,84,85] are called regenerating myofibers. Regenerating myofibers are most commonly defined by the central position of myonuclei in transverse planes of skeletal muscle (Figure 3A). In longitudinal planes of myofibers, central myonuclei are aligned in linear arrays called nuclear chains [86,87,88,89,90,91,92] (Figure 3B). The central position of myonuclei in regenerating myofibers contrasts with the normal, peripheral position of myonuclei near the sarcolemma (Figure 3C) [93,94]. 

The distinction between a mature myotube and a small regenerating myofiber is ambiguous (Table 1). Myotubes in the embryo and in vitro contain 2–20 myonuclei [64,65,66,95] and are qualitatively smaller in length and diameter compared to regenerating myofibers. In contrast, regenerating myofibers contain hundreds of myonuclei [86,90,92], and their radial size when they initially appear in injured muscle is notably smaller than non-regenerating/normal myofibers in the affected muscle (Figure 3A). In contrast to developing myotubes [69,77,82], regenerating myofibers have many organized sarcomeres and myofibrils [40,83,84,87,89,90,91,96]. Developing myotubes and small regenerating myofibers are similar in that they both typically have central myonuclei and express embryonic, fetal, and/or neonatal isoforms of myosin [44,82,96,97,98,99]. The peak expression profile of embryonic myosin corresponds to the emergence of regenerating myofibers after muscle trauma (4–7 days post injury) [38,96,97,98,99,100]. During regeneration, the expression of embryonic and neonatal myosin by myofibers subsides, whereas central myonuclei persist [38,96,97,98,99,100]. Thus, the expression of embryonic myosin by small myofibers during muscle regeneration is commonly interpreted to indicate the presence of myotubes or, more broadly, de novo myofiber formation. On the other hand, a decrease in the expression of embryonic myosin may define the transition of myotubes to regenerating myofibers, or more broadly, an early stage of regenerating myofiber maturation. Suppression of neonatal myosin expression during the course of regeneration is thought to reflect innervation of regenerating myofibers and the transition towards the expression of adult isoforms of myosin [96,100]. 

### 3.4. Stage 4: Regenerating Myofiber Maturation

The final stage of the muscle regeneration paradigm involves the maturation of regenerating myofibers into normal myofibers. In other words, muscle regeneration is thought to be complete when regenerating myofibers are morphologically and functionally indistinguishable from normal myofibers. 

To complete muscle regeneration, regenerating myofibers have to undergo sustained radial growth (hypertrophy) to reach a size that is comparable to normal myofibers as well as relocate a large number of myonuclei from a central to a peripheral position. Regenerating myofibers may also have to undergo myonuclear accretion to support their sustained maturation. In summary, regenerating myofibers have a lot of work to do to become a normal myofiber! 

Despite an extensive body of literature on muscle regeneration, very little is known about the maturation of regenerating myofibers into a normal sized myofiber with peripheral myonuclei. This deficiency in knowledge has perpetuated the belief that regenerating myofiber maturation is analogous to the maturation of myofibers during embryonic and postnatal myogenesis. 

## 4. Embryonic and Postnatal Myogenesis vs. Muscle Regeneration

There is universal agreement that proliferation of muscle progenitor cells and myoblast–myoblast fusion are required for de novo myofiber formation in the embryo, fetus, and during muscle regeneration. Given this agreement, it is often assumed that cellular and molecular processes that facilitate myofiber maturation during muscle regeneration are analogous to those operating in the embryo, fetus, and after birth. This is a reasonable assumption as myogenic cell proliferation, myonuclear accretion, and hypertrophy of myofibers occurs during embryonic myogenesis, postnatal myogenesis, and muscle regeneration. However, myofibers maturing in the embryo, fetus, and after birth are phenotypically different from regenerating myofibers formed after an acute injury. The most striking differences between them are the position, organization, and number of myonuclei. The attainment of a normal myofiber size and morphology is also different during the course of postnatal myogenesis and muscle regeneration. These differences indicate that the cellular and molecular processes that regulate regenerating myofiber maturation are not identical to those that mediate myofiber maturation during embryonic and postnatal myogenesis. 

To provide a framework for delineating cellular and molecular processes that are unique to muscle regeneration, the literature pertaining to indices of myofiber maturation during embryonic myogenesis, postnatal myogenesis, and muscle regeneration in mammals is summarized below. As knowledge of embryonic and postnatal myogenesis provides a foundation for understanding muscle regeneration, myofiber maturation during embryonic and postnatal myogenesis is discussed separately from muscle regeneration. When discussing regenerating myofibers, evidence pertaining to their maturation is discussed in the context of the paradigm of muscle regeneration, and comparisons are made to embryonic myogenesis and/or postnatal myogenesis. 

### 4.1. Myonuclear Positioning during Embryonic and Postnatal Myogenesis

During embryonic myogenesis, myonuclei in developing myotubes/myofibers are dispersed evenly within a relatively small cytoplasmic area that contains isolated sarcomeres and/or developing myofibrils [62,69,77,78,82,95]. Transverse and longitudinal planes of skeletal muscle in the embryo or fetus indicate that myonuclei are centrally located in developing myotubes/myofibers [62,69,77,78,82,95]. The central position of myonuclei persists after myotubes/myofibers are innervated by a motor neuron [101,102]. 

Central myonuclei in myotubes/myofibers in the embryo/fetus move towards the sarcolemma during the course of the development [69,70,77,78]. The time scale of this repositioning, however, is unclear and appears to be temporally associated with myofibrillogenesis [69,77,82,103,104]. This temporal relationship most likely serves as the basis for the widely held belief for myonuclear repositioning during muscle development and regeneration. That is, the formation of myofibrils and function of sarcomeres within them is thought to propel myonuclei from a central to a peripheral position during the course of myofiber maturation. This premise has been supported by evidence of myonuclear trafficking during in vitro myogenesis [105]. Specifically, Roman et al. [105] reported that myofibrillar contraction, as well as desmin-mediated linking of myofibrils together, facilitated the migration of myonuclei towards the sarcolemma. 

The repositioning of myonuclei from a central to a peripheral position during embryonic myogenesis is profound, as few, if any, myofibers at birth have central myonuclei [70,77,106,107,108,109]. Central myonuclei are also not normally observed during postnatal development [78,106,107,108,110,111,112,113,114], which is consistent with the number of myofibers within muscles being fixed at birth or shortly after [28,29,30,31]. The absence of central myonuclei during postnatal development is striking as myofibers rapidly add myonuclei within days of birth and continue to add myonuclei until adulthood [106,107,108,110,111,112,113,114] (discussed below in Section 4.3). 

In summary, central positioning of myonuclei is a phenomenon that is restricted to embryonic myogenesis, as myonuclei are in a peripheral position during postnatal myogenesis. Simply put, repositioning of myonuclei from a central to a peripheral position occurs at a very early stage of myofiber maturation—that is, well before myofibers have reached their adult size and a full complement of myonuclei, myofibrils, organelles, and other components of their cytoplasm. Why myonuclear repositioning occurs at a very early stage of myofiber maturation is unclear. Is the movement of myonuclei from a central to a peripheral position during embryonic myogenesis a consequence of the formation and function of myofibrils? Or does the peripheral positioning of myonuclei prior to birth optimize molecular and cellular processes that facilitate myofiber maturation? If not, why do myonuclei added during postnatal myogenesis take a peripheral, rather than central, position in myofibers? Is their path to a central position during postnatal development blocked by a progressive increase in the number of myofibrils? These fundamentally important questions remain unanswered. 

### 4.2. Myonuclear Positioning during Trauma-Induced Muscle Regeneration

Normally, almost all of the myonuclei within a myofiber are non-randomly dispersed and positioned in a peripheral location near the sarcolemma [93,94] (Figure 3C). In contrast, the majority of myonuclei in regenerating myofibers after chemical or physical trauma are aligned in nuclear chains (Figure 3B) [86,87,88,89,90,91,92,115], which we define as a linear array of at least 10 myonuclei [86,90,92]. Myonuclei are also found near the sarcolemma or clustered together (nuclear clusters) in regenerating myofibers [86,87,88,89,90,91,92]. Nuclear clusters are normally found near the neuromuscular and myotendinous junctions [88,93,116], and a small number of them form outside of these areas in regenerating myofibers [86,88,90,92]. 

Nuclear chains are the most striking, as well as the defining feature of regenerating myofibers. Myonuclei in nuclear chains are very closely aligned and, in many cases, appear to be in contact with each other after chemical or physical trauma in mice [83,86,87,88,89,90,91,115]. In contrast, the majority of central myonuclei during embryonic myogenesis appear to be separated from each other [69,77,82]. The number of myonuclei in nuclear chains of regenerating myofibers is notable, as well as variable. Specifically, the length of individual nuclear chains, a readout of the number of myonuclei in a single nuclear chain, varies considerably (40–90% of myofiber length) and increases during the course of regeneration (Figure 4A) [86]. As discussed later, the length of individual nuclear chains and the total number of nuclear chains in a regenerating myofiber can influence the detection of regenerating myofibers in transverse muscle sections (see “Section 5”). 

Nuclear chains often run in parallel with each other and, in some cases, appear to converge/diverge to produce a ‘Y’ shaped appearance (Figure 4B) [86,89,92,115]. Parallel rows of nuclear chains are indicated in transverse muscle sections by the central position of two or three myonuclei within a myofiber, which is a common occurrence after chemical trauma in rodents (Figure 3A). These organizational patterns of central myonuclei are unique to muscle regeneration, as no discernible rows or organizational patterns of myonuclei have been noted during embryonic myogenesis. 

Using electron microscopy, nuclear chains can be found between myofibrils that contain organized sarcomeres [83,84]. This can also be seen in single myofibers stained to delineate filamentous (F-) actin (Figure 4C) [90] or desmin (Figure 4D), which links Z-lines of adjacent myofibrils together [117]. The position of nuclear chains between myofibrils is also indicated in transverse sections stained with hematoxylin and eosin (Figure 3A) or reagents to detect cytoskeletal proteins (e.g., F-actin or an isoform of myosin) [44,96,98,118]. These techniques reveal that central myonuclei are surrounded by an abundance of cytoskeletal proteins. Similar to regenerating myofibers, central myonuclei during embryonic myogenesis are surrounded by cytoskeletal proteins, including developmental isoforms of myosin [44,82,97]. 

An important aspect of the muscle regeneration paradigm is that central myonuclei move towards the sarcolemma as the radial size of regenerating myofibers increases during the course of regeneration (Figure 2). This tenet likely stems from an apparent temporal relationship between myofibrillogenesis, myofiber hypertrophy, and the repositioning of myonuclei from a central to a peripheral position during embryonic myogenesis [69,77,82,103,104]. Accordingly, the number/percentage of regenerating myofibers is predicted to decrease during the course of muscle regeneration.

A well-recognized phenomenon after chemical and physical trauma to rodents is that regenerating myofibers remain in skeletal muscle for an extended period of time (e.g., 1–6 months post-injury) [14,36,86,87,89,96,119,120,121,122]. Importantly, the extent to which the number/percentage of regenerating myofibers diminishes during the course of trauma-induced muscle regeneration is unclear. For example, the percentage of regenerating myofibers (75–90%) in transverse sections and in isolated myofibers has been reported to remain relatively constant for at least one month of recovery from chemical (barium chloride or notexin) trauma [86,119,122]. Wada et al. [89] reported that the percentage of isolated myofibers that contained nuclear chains (i.e., regenerating myofibers) remained high (~80%) and constant for at least 6 months after physical trauma (400 needle punctures) to skeletal muscle. In contrast, others have reported that the percentage of regenerating myofibers in transverse sections decreases from 3–4 (74–90%) to 60 (14–60%) days of recovery from chemical (bupivacaine or notexin) trauma [14,121]. Conflicting viewpoints and evidence on how long regenerating myofibers remain in skeletal muscle after trauma has created much confusion about myonuclear repositioning during muscle regeneration and its use as an indicator of regenerating myofiber maturation. 

In an effort to gain insight into myonuclear positioning and repositioning during muscle regeneration, we analyzed single myofibers before and after chemical (barium chloride) trauma to gastrocnemius muscles of mice [90]. We found that the percentage of myonuclei in nuclear chains was high at 7 days post-injury (77%, range = 54–94%) and progressively decreased at 14 (70%, range = 60–84%) and 28 (60%, range = 35–80%) days post-injury [90]. The percentage of peripheral myonuclei was low at 7 days post-injury (16%, range = 3–43%) and progressively increased at 14 (24%, range = 12–37%) and 28 (29%, range = 13–55%) days post-injury [90]. Our findings lend credence to the premise that myonuclei move from a central to a peripheral position during the course of muscle regeneration. If true, then the movement of central myonuclei towards the sarcolemma during the course of regeneration is protracted. Because the radial size of regenerating myofibers increased by ~2 fold from 7 to 28 days post-injury [86,90] (discussed below in “Section 6”), the increased prevalence of peripheral myonuclei does not appear to be closely coupled to regenerating myofiber hypertrophy. 

Why regenerating myofibers remain in skeletal muscle for an extended period of time after chemical or physical trauma is unknown. Their sustained presence is either dismissed or interpreted to indicate that muscle regeneration has not yet completed. This interpretation is hard to reconcile with evidence of myofiber size (discussed in “Section 6”), as well as metabolic enzyme activity, mitochondrial respiration, and muscle function returning to control levels within four weeks of chemical trauma [14,15,36,86,96,122,123,124,125]. Maximum specific force of single myofibers (force normalized to myofiber cross-sectional area) after chemical (bupivacaine) trauma has also been reported to be similar to that of control myofibers [126]. Importantly, we are not aware of any study that has demonstrated that the central position of myonuclei or the presence of nuclear chains after muscle trauma impairs specific force or other contractile properties of single myofibers. The extent to which the metabolic profile of regenerating myofibers differs from that of normal myofibers also remains to be determined, as prior research targeting metabolism analyzed regenerating muscles [123,124,125], which normally contains both regenerating and non-regenerating/normal myofibers. 

Myonuclear positioning during muscle regeneration is remarkably different from myonuclear positioning during embryonic and postnatal myogenesis. The alignment and organization of central myonuclei, as well as their sustained presence, are unique features of trauma-induced muscle regeneration. Why myonuclei are uniquely positioned during muscle regeneration is unknown. Is the central positioning of myonuclei during muscle regeneration a consequence of a high rate of myofibrillogenesis and a rapidly expanding cytoplasm? In other words, are myonuclei stuck in a central position because their path towards the periphery is blocked by a large number of myofibrils? Or is the central positioning of myonuclei optimal for orchestrating molecular processes (e.g., transcription) [90,91,92] that facilitate regenerating myofiber maturation? If true, why do central myonuclei persist after regenerating myofibers have reached an adult radial size? Do myonuclei remain in a central position because other aspects of regenerating myofiber maturation have yet to be completed? If yes, what aspect of their maturation lags behind? Clearly, additional research is needed to comprehend why the majority of myonuclei are centrally aligned in regenerating myofibers during trauma-induced muscle regeneration. 

### 4.3. Myonuclear Accretion during Embryonic and Postnatal Myogenesis

Embryonic myogenesis is associated with a high rate of muscle progenitor cell proliferation and myonuclear accretion [61,62,63,70,127,128]. However, the kinetics of myonuclear accretion in developing myotubes/myofibers during embryonic myogenesis has yet to be quantitatively detailed. 

Many studies have evaluated myonuclear accretion during postnatal myogenesis in rodents using myofibers isolated from several different skeletal muscles [106,107,108,110,111,112,113,114,129]. The kinetics of myonuclear accretion was first detailed in extensor digitorum longus (EDL) muscles of mice by White et al. [112]. Strikingly, they reported that the number of myonuclei/myofiber increased by ~3 fold from P3 to P7 [112] (P = postnatal age in days). Myonuclear accretion continued through P21, albeit at a slower rate [112]. Myonuclear accretion observed during the first three weeks of postnatal myogenesis was accompanied by a 3–4 fold increase in both myofiber length and cross-sectional area [112]. 

Others have analyzed myonuclear number in single myofibers isolated from mouse EDL muscles after three weeks of postnatal development [106,108,110,111,114]. These studies demonstrated that myonuclear number, expressed per myofiber or per mm of myofiber length, increased modestly (~10–35%) during the pre-puberty stage of development (P21 to P42) [106,110,111]. Increases in myonuclear number during this period were accompanied by an increase in myofiber length and notable increases in myofiber cross-sectional area [106,108,110,111,112]. Myonuclear accretion appears to continue to some degree after puberty in mice (P42/6 weeks of age) until 8–12 weeks of age [110,130]. During this period, myofiber length reaches adult levels [112,131] and myofiber cross-sectional area continues to increase until adulthood [30,106]. 

As detailed in a narrative review by Bachman et al. [132], the kinetics of myonuclear accretion during postnatal development in mice parallels the number and proliferative state of satellite cells/myoblasts. As a result, variability exists between muscles in myonuclear number, as well as the number of satellite cells associated with a myofiber during the course of postnatal myogenesis [110,112,130,133]. For example, satellite cell and myonuclear number are notably higher in myofibers isolated from soleus compared to EDL muscles throughout postnatal development and in adult mice [93,110,133]. Thus, myonuclear accretion varies between muscles during postnatal development in a manner that appears to reflect the number of myoblasts available for fusion [109]. Furthermore, as bone length increases during postnatal development [134,135] and myonuclear number scales with increases in myofiber length [106,112], differences in myonuclear number between muscles could also reflect differences in architectural (e.g., length and pennation angle) and/or functional features of myofibers [136]. Fundamentally, the genetic and physiological processes that dictate myonuclear number in myofibers during development are not fully understood.

From a physiology perspective, it is conceivable that increases in bone length drive the longitudinal growth of myofibers during postnatal development, which in turn creates a need for additional myonuclei to sustain both longitudinal and radial growth of myofibers. Upon myofibers reaching an adult length, the need for myonuclear accretion to sustain radial growth of myofibers could stem from increases in body mass and muscle use. Deciphering the cellular and molecular interplay between developing bone, satellite cells, and the needs of myofibers during postnatal myogenesis will undoubtedly be challenging. Equally challenging will be to determine why myoblasts during postnatal myogenesis fuse with existing myofibers and not with each other to form myotubes/myofibers. This is a fundamentally important question as myofiber number in a given muscle is widely believed to be fixed around the time of birth [28,29,30,31]. 

### 4.4. Myonuclear Accretion during Trauma-Induced Muscle Regeneration

The paradigm for muscle regeneration proposes that myonuclear accretion in myotubes facilitates their longitudinal and radial growth, which in turn gives rise to small regenerating myofibers (Figure 2). Although the distinction between a mature myotube and a small regenerating myofiber is ambiguous (discussed above in “Section 3.3”), myonuclear accretion is thought to continue until regenerating myofibers obtain the normal complement of myonuclei. The attainment of a normal number of myonuclei is believed to be necessary for regenerating myofibers to achieve a radial size that is comparable to a normal myofiber. Surprisingly, little research has directed towards detailing myonuclear accretion during regeneration [86,90,92].

We have evaluated myonuclear accretion during muscle regeneration by analyzing single myofibers before and after chemical (barium chloride) trauma to tibialis anterior and gastrocnemius muscles of mice [86,90,92]. The magnitude and rate of myonuclear accretion observed during the course of muscle regeneration was surprising. Specifically, the number of myonuclei in regenerating myofibers, expressed per unit myofiber length (myonuclei/100 μm), was ~2 fold higher at 7, 14, and 28 days post-injury compared to non-regenerating myofibers in control muscles (0 days post-injury) [86,90]. Furthermore, myonuclear number (myonuclei/100 μm) in regenerating myofibers was similar at 7, 14, and 28 days post-injury [86,90]. The magnitude and rate of myonuclear accretion that we observed in regenerating myofibers paralleled the robust satellite cell/myoblast proliferation that occurs within 7 days of chemical trauma in mice [36,37,38]. The profile of myonuclear accretion, however, was not temporally related to regenerating myofiber hypertrophy, as cross-sectional area of regenerating myofibers increased by 2.2 fold from 7 to 14 days post-injury and by 17% from 14 to 28 days post-injury [90] (discussed below in “Section 6”). Simply put, myonuclear accretion preceded the robust hypertrophy of regenerating myofibers. Our findings demonstrate that regenerating myofibers obtain a higher-than-normal number of myonuclei within 7 days of chemical (barium chloride) trauma [86,90]. 

Although reporting the myonuclear number relative to myofiber length (e.g., myonuclei/100 μm) is informative and widely done, it is not a measure of the absolute number of myonuclei in a myofiber. This is an important issue to consider when myofiber length is changing, such as during postnatal myogenesis [112,131]. Unfortunately, changes in the length of myotubes or regenerating myofibers during regeneration have not been described in the literature. The measurement of myotube/regenerating myofiber length requires that they remain intact after being enzymatically and physically liberated from a whole muscle, procedures that can fracture small, fragile myofibers. Nevertheless, further work is needed to detail changes in regenerating myofiber length and the absolute number of myonuclei in regenerating myofibers to fully appreciate the kinetics of myonuclear accretion during muscle regeneration. 

Why regenerating myofibers rapidly add more than the normal complement of myonuclei is unknown. Intuitively, the number of myonuclei in a regenerating myofiber reflects both extrinsic and intrinsic regulation of myonuclear accretion. Extrinsic factors, such as molecules (e.g., cytokines) and cell types (e.g., macrophages) within injured muscle, increase the number of myoblasts available for fusion by augmenting satellite cell/myoblast proliferation [23,26,32,33]. The local environment of injured muscle could also enhance the ability of myoblast to fuse to myotubes/regenerating myofibers by augmenting the expression and/or function of molecules that facilitate cell-to-cell adhesion [25,51,52] and/or membrane dynamics during fusion [24,53,54]. Independent of the extrinsic regulation of myonuclear accretion, myotubes/regenerating myofibers likely have an intrinsic need to add myonuclei to support their sustained maturation. However, such a need could only be satisfied if satellite cells proliferate, and myoblasts are available for fusion. 

To reconcile the above logic, we propose that myotubes/regenerating myofibers obtain a sufficient number of myonuclei to sustain their maturation through an exquisite interplay between their needs and the local environment of injured muscle. The details of such an interplay are difficult to fathom as they likely involve paracrine signaling in myotubes/regenerating myofibers, enhanced function of molecules that mediate cell-to-cell adhesion and fusion, and synergistic interactions with multiples cytokines and cell types. Alternatively, the myonuclear number in regenerating myofibers could be solely dependent on the extrinsic regulation of satellite cell/myoblast proliferation and fusion. Such a scenario is not favored because it implies that developing myotubes/regenerating myofibers after muscle trauma have no role in determining how many myonuclei they house. In other words, it does not seem prudent to ignore the intrinsic needs of myotubes/regenerating myofibers, particularly when their longitudinal and radial growth demands an increase in transcription. 

Given the proposed regulation of myonuclear number in regenerating myofibers, we speculate that the higher-than-normal number of myonuclei in regenerating myofibers, which was obtained within 7 days of muscle trauma [90,92], is a consequence of extrinsic regulation of myogenic cell proliferation and fusion. In other words, we think that the pro-proliferative and pro-fusion environment of severely injured muscle caused the myonuclear number to exceed levels that are required for regenerating myofiber maturation. Our hypothesis for why myonuclear number in regenerating myofibers is higher-than-normal does not negate the possibility that myonuclear accretion in myotubes/regenerating myofibers after muscle trauma is need-dependent. Rather, we think that extrinsic factors after muscle trauma overwhelm the intrinsic need for regenerating myofibers to add myonuclei. 

Superficially, myonuclear accretion during muscle regeneration is different from that occurring during postnatal myogenesis. Regenerating myofibers obtain a higher-than-normal number of myonuclei within 7 days of muscle trauma [90,92]; whereas myofibers developing after birth take at least a couple of months to obtain adult levels of myonuclei [106,110,130]. As cells of the myogenic lineage (e.g., myoblasts) are the principal source of myonuclei for all myofibers [37,43,44,45], it is likely that differences in the magnitude and kinetics of myonuclear accretion between muscle regeneration and postnatal myogenesis parallel differences in satellite cell/myoblast proliferation. Specifically, we suspect that satellite cell/myoblast proliferation after chemical trauma is notably higher for a shorter period of time compared to postnatal myogenesis. If true, this could reflect a differential profile of cell types (e.g., macrophages) and cytokines during muscle regeneration and postnatal myogenesis. This speculation is reasonable as a large number of macrophages accumulate in injured muscle and their secretome augments myogenic cell proliferation after chemical trauma [26,33,137]. However, the macrophage accumulation profile during postnatal myogenesis, as well as the extent to which their secretome contributes to myogenic cell proliferation and/or fusion during postnatal myogenesis, remains to be determined. Independent of the extrinsic regulation of myonuclear accretion, it is also possible that myotubes/regenerating myofibers have an intrinsic need to quickly obtain a large number of myonuclei, a need that may not be germane to postnatal myogenesis. 

### 4.5. Myofiber Hypertrophy during Postnatal Myogenesis

Myofibers at birth undergo sustained hypertrophy until adulthood. This hypertrophy has been attributed to a progressive increase in the number and size of myofibrils, which normally occupy nearly all of the cytoplasm of myofibers [103,104]. Because sustained hypertrophy during postnatal myogenesis demands a net accumulation of myofibrillar proteins [138,139], there has been a long standing interest in deciphering the contribution of myonuclear accretion and downstream processes (e.g., transcription and translation) to myofiber hypertrophy during postnatal development. 

Prior research has reported statistically significant positive correlations between myonuclear number and myofiber cross-sectional area or volume at selected time points of postnatal development [108,110,112,129]. These findings indicated a role of myonuclear accretion in facilitating myofiber hypertrophy during postnatal myogenesis.

Two studies have directly tested the relationship between myonuclear number and the radial size of myofibers during postnatal myogenesis in mice [106,110]. Cramer et al. [106] inhibited myoblast fusion by conditionally deleting *myomaker* during the first, second, or third week of postnatal development in mice [106], which corresponded to a timeframe when myofibers are rapidly adding myonuclei [112]. Deleting *myomaker* during the first of week of postnatal development substantially reduced myonuclear number and myofiber volume in myofibers isolated from EDL muscles at 4 weeks of postnatal development [106]. Notable reductions in myofiber cross-sectional area were also observed in other hindlimb muscles at 4 weeks of postnatal development [106]. Interestingly, deleting *myomaker* during the first of week of postnatal development resulted in premature death, which was interpreted to indicate a finite ability of myonuclei to compensate for severe impairments in myonuclear accretion [106]. Deleting *myomaker* during the second week of postnatal development reduced myonuclear number in EDL myofibers by ~35–55% throughout 5 months of postnatal development [106]. Importantly, sustained reductions in myonuclei number did not impair the ability of myofibers to undergo hypertrophy. Rather, sustained reductions in myonuclei number prevented myofibers from reaching a radial size that is comparable to controls [106]. Bachman et al. [110] also reported that reductions in myonuclear number prevented myofibers from reaching a normal size during postnatal myogenesis. This was demonstrated by depleting satellite cells (Pax7+ cells) after 4 weeks of postnatal development in mice. The near absence of satellite cells in EDL and soleus muscles reduced myonuclear number by 10–15% and myofiber cross-sectional area by 15–19% at 8 weeks of postnatal development [110]. The findings of Cramer et al. [106] and Bachman et al. [110] demonstrate that myonuclear number is a critical determinant of the radial size of myofibers during postnatal myogenesis. In other words, if the intrinsic need for additional myonuclei during postnatal myogenesis is not met via myogenic cell proliferation and subsequent fusion, myofibers fail to reach their normal radial size [106,110]. 

Additional research is needed to understand how the need for myofibrillar protein synthesis to sustain myofiber hypertrophy during postnatal myogenesis is coordinated amongst the myonuclei, and how such a need triggers myonuclear accretion. This will be a challenging endeavor as the transcriptome of individual myonuclei during postnatal myogenesis is heterogeneous [140] and the translational regulation of protein synthesis is complex [141,142]. Interestingly, Cramer et al. [106] proposed that individual myonuclei have a finite transcriptional capacity during postnatal myogenesis. If true, myonuclear accretion during postnatal myogenesis could be initiated when individual myonuclei are approaching their maximal transcriptional capacity. 

### 4.6. Regenerating Myofiber Hypertrophy after Trauma-Induced Muscle Regeneration

According to the muscle regeneration paradigm, regenerating myofibers undergo hypertrophy until they reach a radial size that is comparable to normal myofibers (Figure 2). Furthermore, the radial size of a regenerating myofiber is thought to be closely coupled to myonuclear number. The logic behind the coupling is that myonuclear accretion increases the capacity of regenerating myofibers to transcribe genes, which, in theory, optimizes their hypertrophy. The transcriptional activity in regenerating myofibers and the division of transcriptional labor amongst myonuclei within them is poorly understood [90,91,92,143].

A large number of studies have quantified the radial size of myofibers before and after muscle trauma in rodents. It is difficult to construct a composite profile of regenerating myofiber hypertrophy after trauma for several reasons. One, temporal variability in indices of muscle regeneration exists between experimental models of inducing muscle trauma [36,144]. Two, the radial size of myofibers after trauma is routinely measured at one or two recovery time points. Lastly, in some cases, it is unclear if the reported data after trauma is representative of all myofibers (non-regenerating and regenerating) or just regenerating myofibers. The reason for raising this issue is not to question the appropriateness of data reporting in prior studies, as there are valid reasons for reporting the radial size of all myofibers after trauma. Rather, the intent is to highlight our belief that regenerating myofiber hypertrophy cannot be accurately described from changes in the radial size of all myofibers. This belief is rooted in the fact that the number/percentage of regenerating myofibers in transverse sections after chemical or physical trauma can vary between muscles at any recovery time point (e.g., 46–93%) [86]. This variability means that the number/percentage of non-regenerating myofibers in transverse sections after muscle trauma also varies at each recovery time point (e.g., 7–54%) [86]. A secondary analysis of published data [86] revealed that the cross-sectional area of non-regenerating myofibers after chemical (barium chloride) trauma is 30–55% lower compared to non-regenerating myofibers in control muscle (Figure 5). This analysis supports our belief that drawing interpretations on regenerating myofiber hypertrophy from values that are representative of all myofibers can be confounded by the presence of non-regenerating myofibers after muscle trauma. To address all the challenges mentioned while providing insight into the kinetics of regenerating myofiber hypertrophy, we highlight evidence pertaining to changes in the radial size of regenerating myofibers during the course of regeneration. 

Although regenerating myofibers are discernible 4 days after chemical trauma in mice (Figure 2), it is difficult to accurately measure their radial size in transverse muscle sections and to isolate them from whole muscles. On the other hand, regenerating myofibers at 7 days post-injury and beyond can be measured in transverse sections, as well as isolated from whole muscles of mice. We found that regenerating myofiber cross-sectional area in transverse sections (Figure 5) [86] and in isolation [90] increased by 1.7–2.6 fold between 7 and 14 days after chemical (barium chloride) trauma in mice. Others have also reported a robust hypertrophy of regenerating myofibers over a similar time frame of recovery from chemical (barium chloride or notexin) trauma in mice [96,99,145,146]. Rapid hypertrophy during early recovery from chemical trauma is preceded by myonuclear accretion [90] and is temporally associated with innervation of regenerating myofibers [96,120] and a high rate of muscle protein synthesis [86]. Regenerating myofiber cross-sectional area continued to increase between 14 and 21 days post-injury, albeit at a slower rate [145] (Figure 5). This slower rate of hypertrophy was temporally associated with a gradual return of protein synthesis rates to levels that were observed in control muscles [86]. No further increase in regenerating myofiber cross-sectional area was observed between 21 and 28 days post-injury (Figure 5). Interestingly, cross-sectional area of regenerating myofibers at 14 days post-injury was comparable to the size of normal (non-regenerating) myofibers in control muscles (Figure 5). Despite reaching control levels, regenerating myofiber cross-sectional area continued to increase to levels that were 25% and 18% higher than control myofibers at 21 and 28 days post-injury, respectively (Figure 5). Others have reported that the radial size of myofibers exceeded control levels within 4 weeks of chemical (notexin) trauma [122] and remained elevated for at least 6 months after chemical (cardiotoxin, barium chloride, and notexin) or physical (cold exposure) trauma to skeletal muscle [36]. The hypertrophy of regenerating myofibers beyond control levels is not consistent with a central tenet of the muscle regeneration paradigm. That is, hypertrophy of regenerating myofibers is thought to cease when they achieve a size that is comparable to normal myofibers.

We have examined the statistical relationship between myonuclear number and the radial size of regenerating myofibers after chemical (barium chloride) trauma in mice [86,90,92]. Significant correlations were observed between myonuclear number and regenerating myofiber area (r = 0.67) [86] and volume (r = 0.77) [90]. Interestingly, regenerating myofiber volume was closely coupled to myonuclei number at 7 days post-injury (r = 0.95) and became more responsive to myonuclear number at 14 and 28 days post-injury, as indicated by a progressive increase in the slope of regression lines during the course of regeneration [90]. The increased responsiveness of myofiber volume to myonuclear number was temporally associated with an elevated level of transcription within regenerating myofibers [90]. Taken together, our results are consistent with the premise that myonuclear number and their transcriptional activity are predictors of the radial size of regenerating myofibers after muscle trauma. 

No studies have directly tested the relationship between myonuclear number and the radial size of regenerating myofibers during regeneration. In contrast, prior studies have tested the contribution of myonuclear accretion to myofiber hypertrophy during postnatal myogenesis [106,110] and after resistance exercise [147,148] and muscle overload (synergist ablation) [47,149,150]. To be clear, investigators have demonstrated that genetically depleting satellite cells [38,44,46,47,48], *myomaker* [60], or *myomixer* [56] prior to chemical trauma in mice prevents regenerating myofiber formation during recovery. Because few, if any, regenerating myofibers formed in the absence of satellite cells, *myomaker*, or *myomixer*, these studies [38,44,46,47,48,56,60] do not provide insight into the relationship between myonuclear number and the radial size of regenerating myofiber during regeneration. Unfortunately, current genetic models are not well suited to test such a relationship as they require several days of Cre-mediated recombination to produce the desired outcome (e.g., depletion of satellite cells or absence of myomaker). This is problematic because myonuclear number in regenerating myofibers reaches a plateau within 7 days of muscle trauma [86,90,92]. In other words, initiating Cre-mediated recombination just prior to or immediately after muscle trauma may not reduce the number of satellite cells or myomaker expressing myoblasts to a level that compromises myonuclear accretion during recovery. Thus, testing the extent to which myonuclear number dictates the radial size of regenerating myofibers may require the development of new approaches that manipulate myonuclear number or myonuclear function in regenerating myofibers during the course of regeneration. 

Regenerating myofibers during muscle regeneration and myofibers during postnatal myogenesis in mice are similar in that they both undergo robust hypertrophy. However, regenerating and developing myofibers differ in the time needed to reach an adult size, as well as the temporal relationship between myonuclear accretion and myofiber hypertrophy. Regenerating myofibers achieve an adult radial size much sooner (3–4 weeks) [36,86,90,122] than developing myofibers (8–12 weeks) [30,106]. During muscle regeneration, myonuclear accretion reaches a plateau within 7 days of chemical trauma and precedes notable increases in the radial size of regenerating myofibers [86,90]. In contrast, myonuclear accretion and myofiber hypertrophy during postnatal myogenesis are closely coupled, particularly when myofibers are rapidly undergoing hypertrophy [110,112]. Reconciling such disparities awaits a better understanding of the extrinsic and intrinsic regulators of myonuclear accretion and myofiber hypertrophy during muscle regeneration, as well as during postnatal myogenesis. 

### 4.7. Myofiber Formation and Morphology during Embryonic Myogenesis

It is widely accepted that myofiber formation during embryonic myogenesis in mammals occurs in two successive stages [69,70,77,78]. Each stage has been attributed to the proliferation and fusion of distinct populations of progenitor cells, namely embryonic and fetal myoblasts [151,152,153]. Morphological, functional, and molecular differences between embryonic myoblasts, fetal myoblasts, and satellite cells have been extensively studied [154,155,156,157], as well as summarized in narrative reviews [151,152,153]. 

The first stage of embryonic myogenesis in mice (E10.5–12.5, E = embryo age in days) produces ‘primary’ myotubes from the fusion of embryonic myoblasts [69,70,77]. Primary myotubes form independent of innervation, accumulate myonuclei, span tendon-to-tendon, and serve as the structural framework for the second stage of embryonic myogenesis [61,62,70,77,101,158,159,160]. The second stage in mice (E14.5–17.5) gives rise to ‘secondary’ myotubes as a result of fusion of fetal myoblasts [70,158]. Several secondary myotubes form within the basal lamina of primary myotubes and near their mid-belly [77,78] (Figure 6A). Secondary myotube formation is dependent on innervation of primary myotubes, as well as muscle contractions [101,158,161]. 

The maturation of secondary myotubes is a critical step in embryonic myogenesis as they give rise to ~80% of the myofibers in adult muscle [101,160]. The maturation of secondary myotubes is characterized by innervation, myonuclear accretion, myofibrillogenesis, longitudinal and radial growth, and the assembly of a basal lamina [61,62,63,69,77,78,162]. Observations made via electron microscopy during the course of embryonic myogenesis indicate that myotubes either fuse with each other [69,70] or separate [77,78,162]. Prior investigators have emphasized and schematically depicted that primary and secondary myotubes can appear in a transverse section as a single cell, a cell undergoing longitudinal splitting, or two distinct cells during the course of embryonic myogenesis (Figure 6B) [69,70,160,162,163]. In the latter case, secondary myotubes were distinguished from primary myotubes/myofibers based on their smaller radial size [70,77,78,101,158,162]. Because secondary myotubes appear to separate from primary myotubes/myofibers throughout the course of embryonic myogenesis, investigators have noted the difficulty in quantifying myofiber number in transverse sections around the time of birth [30,78,162,163]. 

### 4.8. Myofiber Formation and Morphology during Trauma-Induced Muscle Regeneration

De novo myofiber formation after chemical trauma in mice is dependent on satellite cells (Pax7+ cells) [38,44,46,47,48]. Although several types of progenitor cells accumulate in skeletal muscle after trauma (e.g., Pax3+, fibro-adipogenic progenitors, Twist2+, mesoangioblasts, and PW1+ cells) [23,164,165], they are unable to fully compensate for the near absence of Pax7+ satellite cells [38,44,46,47,48]. This indicates that Pax7+ satellite cells are the principal, if not the sole precursor cells that gives rise to regenerating myofibers in adult mice after trauma. In contrast, de novo myofiber formation during embryonic myogenesis has been attributed to different progenitor cells, namely embryonic and fetal myoblasts [152,157]. 

A well-recognized phenomenon during muscle regeneration is that some myofibers appear to be either undergoing longitudinal splitting or fusion. In transverse sections, a ‘split’ or fissure within some myofibers can be observed in histologically stained sections (e.g., hematoxylin and eosin) [40,145,166], as well as sections treated to delineate the sarcolemma or basal lamina [167] (Figure 7A). Griffin et al. [145] analyzed serial transverse sections and reported that 20–40% of the myofibers exhibited a ‘split’ morphology 5 days after chemical (barium chloride) trauma in mice. The incidence of ‘split’ myofibers during prolonged recovery from chemical trauma remains to be determined.

Myofibers isolated after trauma also exhibit an unusual morphology called branching [40,86,119,145,168]. A branched myofiber is defined as an isolated myofiber that has a membrane that is contiguous with one or more myofibers. Characteristics of a branched myofiber include a ‘split’/fissure typically near the middle of a myofiber (Figure 7B) or a bifurcation/process on their side or at their end [40,86,119,145,168]. Given the characteristics of myofiber branching [40,86,119,145,168], it is understandable how an individual branched myofiber after trauma could appear in transverse planes as a single cell, a cell undergoing longitudinal splitting or fusion, or two cells (Figure 7C).

Myofiber branching is restricted to regenerating myofibers, as very few non-regenerating myofibers after chemical (barium chloride and cardiotoxin) trauma in mice showed signs of branching [119]. A moderate to high percentage of regenerating myofibers (25–60%) showed signs of branching 10–21 days after chemical trauma in mice [86,119,145,146]. The percentage of branched regenerating myofibers remains elevated (28–60%) during prolonged recovery (1 to 6 months) from chemical trauma in mice [86,119]. 

Myofiber ‘splitting’ and/or branching also occurs in hypertrophying [169,170] and diseased muscle (e.g., in mdx mice) [95,119,171,172], as well as a result of aging [119]. Although each condition in which ‘split’ and/or branched myofibers are observed is associated with muscle regeneration to some degree, the environment and the phenotype of regenerating, hypertrophying, and diseased muscle is distinctly different. This has resulted in wide speculation on the cause and physiological significance of ‘split’ and/or branched myofibers. A comprehensive overview of the abnormal morphology of some myofibers in the context of muscle hypertrophy [169,170] and disease [172] can be found elsewhere. 

In trauma models of muscle regeneration, ‘split’/branched myofibers are thought to represent active/incomplete fusion of myotubes/myofibers to each other [86,119,168,173]. This premise is supported by qualitative observations of myotube/myofiber fusion after muscle trauma [40,83,84,85]. Furthermore, molecules expressed by myogenic cells that facilitate or augment their fusion in vitro have also been reported to influence myofiber branching after chemical (barium chloride) trauma in mice [86,145,146]. The sustained presence of branched myofibers after trauma is believed to reflect a protracted or stalled fusion, as well as to confound interpretations of an increased number (hyperplasia) of myofibers after chemical trauma [40,122,166]. Specifically, some believe that myofiber branching produces a pseudo hyperplasia when myofiber number is quantified in transverse sections. 

When considering the morphology of myofibers during embryonic myogenesis (Figure 6), another possibility exists for the formation of ‘split’/branched myofibers during muscle regeneration. That is, the unusual myofiber morphology could reflect myotubes/regenerating myofibers that are separating from each other during the course of regeneration. To be clear, this scenario is distinctly different from the premise that extreme myofiber hypertrophy causes longitudinal splitting of the affected myofiber [169,170]. Prior investigators have reported that myotubes form within and outside of the basal lamina of necrotic myofibers and that they assemble their own basal lamina after chemical (barium chloride) and physical (e.g., cold exposure and crush) trauma [37,39,40,41,167]. Schmalbruch et al. [40] noted that “usually, one basal lamina tube contained several myotubes” after physical trauma. The assembly of a basal lamina by myotubes/regenerating myofibers after trauma is thought to ultimately replace the basal lamina of necrotic myofibers [37,39,40,41,167]. The extent to which myotubes/regenerating myofibers assemble their own basal lamina could influence the degree to which they fuse or separate from each other, as the basal lamina would likely serve as a physical barrier for their fusion. Nevertheless, closely aligned myotubes/regenerating myofibers could appear as ‘split’/branched myofibers if they are separating from each other during the course of regeneration. In this scenario, the sustained presence of ‘split’/branched myofibers during prolonged recovery could reflect a protracted or stalled assembly of a basal lamina for each myofiber. 

The possibility that myofiber ‘splitting’ and branching during regeneration reflects the separation of myotubes/regenerating myofibers is intriguing for two reasons. One, the morphology of ‘split’/branched myofibers during regeneration resembles the morphology of primary and secondary myotubes/myofibers separating from each other during the course of embryonic myogenesis (Figure 6B). However, only the ‘split’ morphology has been noted during muscle development [69,70,101,162,163], as no study has examined isolated myotubes/myofibers for signs of branching during embryonic myogenesis. Thus, the extent to which myotubes/myofibers that are separating from each other would appear as a branched myofiber after being liberated from embryonic muscle is unknown. Furthermore, myotubes/myofibers formed during embryonic myogenesis assemble their own basal lamina [77,174,175], whereas their formation during trauma-induced muscle regeneration primarily occurs within existing basal lamina (i.e., the basal lamina of necrotic myofibers) [37,39,40,41,167]. This difference further complicates the extrapolation for findings from embryonic myogenesis to trauma-induced muscle regeneration. The premise that myotube/regenerating myofiber separation explains myofiber ‘splitting’/branching during regeneration is also intriguing because we have noticed that the total number of regenerating myofibers in transverse sections increases markedly from 7 to 28 days after chemical (barium chloride) trauma in mice. This increase could be attributable to a pseudo hyperplasia resulting from incomplete myotube/myofiber fusion, myofibers separating from each other during the course of regeneration, or sustained de novo myotube/regenerating myofiber formation. The latter possibility seems unlikely as satellite cell/myoblast proliferation and markers of de novo myofiber formation (e.g., embryonic myosin heavy chain expression) are substantially reduced or non-existent after 7 days of recovery from chemical trauma in mice [36,37,38,96,97,98]. Nevertheless, the underlying processes that cause myofiber ‘splitting’ or branching after trauma-induced muscle regeneration remain to be determined.

## 5. Complete vs. Segmental Necrosis of Myofibers and Ensuing Muscle Regeneration

In the traditional sense, myofiber necrosis implies death of the entire myofiber (complete necrosis), which is difficult to unambiguously establish. Myofiber necrosis is routinely indicated in transverse sections by the accumulation of nuclei/cells in the cytoplasm (indicative of sarcolemma discontinuity or rupture) and a fragmented cytoplasm (indicative of extensive cytoskeletal disruptions). Myofiber necrosis is also indicated by a quantitative [14,176,177,178,179] or qualitative [38,48] loss in the number of myofibers in transverse sections within days of the injury. On the other hand, myofiber injury in transverse sections is indicated by a pale cytoplasmic staining (indicative of cytoskeletal disruptions or loss of cytoskeletal proteins) and a swollen/rounded appearance (indicative of edema). 

A common practice in muscle injury research is not to quantitatively distinguish between myofiber necrosis and injury, whereas muscle regeneration research does not normally include a quantitative analysis of myofiber necrosis or injury. These practices, in conjunction with the ambiguity in defining myofiber necrosis, make it difficult to objectively determine the relationship between myofiber necrosis and regenerating myofiber formation between experimental models of muscle injury and regeneration. 

It is widely accepted that intramuscular injection of a chemical toxin (e.g., cardiotoxin, barium chloride, notexin, or bupivacaine) causes complete necrosis of a large number of myofibers within days of the injection [14,38,48]. On the other hand, prior work using light and electron microscopy have demonstrated that localized physical trauma to skeletal muscle through puncture, crush, or laceration produces signs of necrosis in a limited area of a myofiber [40,85,87,115,173,180,181,182,183,184]. This is commonly referred to as ‘segmental or focal necrosis’ as the sarcolemma and the ultrastructure of the affected myofiber appear to be intact in regions proximal to the focal area of damage [85,115,173,180,182,185]. Segmental necrosis of myofibers also occurs when a small or a large portion of a skeletal muscle has been surgically removed (e.g., volumetric muscle loss models), as myofibers are still evident in the vicinity of the focal lesion during recovery [5,186,187,188,189].

Similar to complete necrosis, segmental necrosis of myofibers initiates processes associated with muscle regeneration (Figure 8). Prior work has demonstrated that myoblasts and other cell types accumulate in the necrotic zone, and that myotubes with central myonuclei fill the void between the stump ends of myofibers exhibiting segmental necrosis [85,87,115,173,180,185,190]. The extent to which myotubes formed in the localized region express embryonic, fetal, and/or neonatal isoforms of myosin remains to be determined. Myotube maturation in the localized region is thought to restore continuity and structure to the affected myofiber [85,115,173,180,182,185]. In this way, myogenic cell proliferation and myotube formation in response to segmental necrosis should be considered processes of ‘myofiber repair’. Importantly, segmental necrosis of myofibers does not always result in complete repair as the void created by laceration can be filled with fibrotic tissue [181,183,184,191]. Surgical removal of a relatively large portion of a skeletal muscle is also accompanied by fibrosis, as well as the near absence of regeneration during recovery [5,186,187,188,189]. In this case, the impaired regeneration is attributable to the removal of satellite cells and the extracellular matrix from the affected area [5,186,187,188,189]. 

Common usage of ‘muscle regeneration’ does not explicitly distinguish between complete or segmental necrosis of myofibers, which can lead to confusion. For example, many narrative reviews illustrate muscle regeneration in response to segmental necrosis of myofibers (Figure 8) [17,18,55]; whereas other reviews illustrate muscle regeneration resulting from complete necrosis of myofibers (Figure 2) [18]. This issue may seem trivial as complete and segmental necrosis of myofibers both elicit processes of muscle regeneration. However, the distinction between complete and segmental necrosis of myofibers becomes important when quantifying regenerating myofibers in transverse sections. 

As mentioned above (see “Section 4.2”), individual nuclear chains can span 40–90% of the measured length of an isolated regenerating myofiber [86]. Given the length of individual nuclear chains (Figure 4A) [86] and a qualitative assessment of the number of nuclear chains in a regenerating myofiber, we estimate that central myonuclei occupy at least 80% of the length of regenerating myofibers after chemical (barium chloride) trauma in mice [86,90,92]. This means that at least one central myonucleus would be found in nearly all transverse sections of an individual regenerating myofiber after chemical trauma (Figure 9A), which would be predicted to occur if myofibers underwent complete necrosis prior to regenerating myofiber formation. On the other hand, central myonuclei are found in the region of a myofiber that exhibited segmental necrosis [85,115,173,180,182,185]. Depending on the extent and location of the segmental necrosis, the localized region of repair could be variable in length and located anywhere along the length of a myofiber. Thus, myofiber repair in response to segmental necrosis would be expected to produce at least one central myonucleus in a limited number of transverse planes of the affected myofiber (Figure 9B). This would result in large variability in the quantitative analysis of regenerating myofibers in transverse sections of a muscle.

The above evidence and logic are intended to substantiate our belief that detection of regenerating myofibers in transverse sections is profoundly impacted by the cause of the muscle injury (e.g., chemical toxin, localized physical trauma, or injurious contractions), which in turn dictates the degree to which myofibers undergo complete or segmental necrosis. This should be considered when trying to reconcile quantitative analysis of regenerating myofiber formation and maturation within and between experimental models, particularly in models that are not expected to cause complete necrosis of myofibers. 

It is important to acknowledge that we have infrequently encountered regenerating myofibers after chemical (barium chloride) trauma in mice that contained central myonuclei/nuclear chains in a localized region, a region that was less than 20% of the length of the myofiber. In such rare occurrences (~1% of the regenerating myofibers analyzed), the majority of myonuclei in the regenerating myofiber were in a peripheral, not central, position. Our analysis of regenerating myofibers after chemical trauma in mice indicates that nearly all of them reflect muscle regeneration in response to complete necrosis of myofibers and that a very small number of them reflect segmental necrosis. 

## 6. Regenerating Myofibers in Models of Contraction- and Exercise-Induced Muscle Injury

### 6.1. Animal Models

Regenerating myofibers in animals have been reported in some, but not all, experimental models of contraction- or exercise-induced muscle injury. The variance between the models most likely reflects the extent to which the experimental approach caused myofiber necrosis (complete and/or segmental), which is a prerequisite for regenerating myofiber formation in injured muscle [17]. 

Animal models consisting of a high number of nerve-stimulated forced lengthening (or eccentric) contractions consistently result in overt signs of myofiber injury/necrosis, as well as a sustained deficit in muscle function [176,177,178,179,192,193,194,195,196]. Investigators have noted after lengthening contractions in mice that “many fibres that appear normal in one histological section show signs of injury in another section” [176] and that “some fibers had multiple areas of disruption and normal areas in between” [177]. These comments, in conjunction with observations made via electron microscopy [196], indicate that segmental injury/necrosis of myofibers is a prevalent feature of contraction-induced muscle injury. Quantitatively, contraction-induced myofiber necrosis (complete or segmental) in transverse sections is indicated by a modest increase in the number/percentage of myofibers that were invaded by cells [192,197], as well as a 20–50% decrease in the number of myofibers at 3 days of recovery [176,177,178,179]. Other animal models such as synergist ablation, downhill running, and muscle reloading produce ultrastructural signs of focal injury to myofibers [6,198,199,200,201,202]. However, in these models, few, if any, myofibers display overt signs of myofiber necrosis [200,201,203,204,205,206,207]. 

Prior research that has reported signs of myofiber necrosis after lengthening contractions have also reported the presence of regenerating myofibers during recovery (Table 2) [177,179,192,193,194,197,208,209,210,211,212]. The time scale of regenerating myofiber formation and maturation after injurious contractions is difficult to discern for multiple reasons. Namely, no single study has analyzed regenerating myofibers in transverse sections or in isolation, throughout prolonged recovery from lengthening contractions. Nevertheless, it appears that regenerating myofibers gradually emerge within 2 weeks of recovery from injurious lengthening contractions [192,193,194] and can remain in skeletal muscle for at least 2 months [179,208,210,212]. The percentage of regenerating myofibers in transverse sections has been reported to be highest in the mid-belly (~40%) compared to the ends (<10%) of muscles 14 days after injurious contractions [193]. Hypertrophy of regenerating myofibers after injurious contractions is largely unknown as only one study reported that the radial size of regenerating myofibers increased by ~20% from 7 to 14 days of recovery [192]. Overload of the plantaris (via surgical removal of soleus and gastrocnemius muscles; classic synergist ablation model) has been reported to increase the prevalence of regenerating myofibers at 1 (28–35%) and 2 (8–30%) weeks of overload [47,149,206,213]. In other animal models of contraction- or exercise-induced muscle injury (e.g., muscle reloading and downhill running), few, if any, regenerating myofibers were observed during recovery [199,203,214,215]. 

De novo myofiber formation after injurious contractions has also been evaluated through myofiber expression of embryonic or developmental myosin. The percentage of myofibers expressing embryonic myosin after lengthening contractions has been reported to be low (~2%) in mice [192] and to be elevated at 3 and 7 (~10–20%), but not 28 days of recovery in rabbits [195]. Others reported that lengthening contractions in rodents caused the absolute number of myofibers expressing embryonic or developmental myosin to progressively increase for up to 7–10 days of recovery [197,216]. These findings are consistent with a gradual rise in regenerating myofibers after injurious contractions [192,193,194].

### 6.2. Human Models

There is no doubt that strenuous physical activity in humans can cause signs (e.g., increased blood levels of creatine kinase and focal lesions in the cytoskeleton of myofibers) and symptoms (e.g., soreness and muscle dysfunction) of a muscle injury [2]. What remains in doubt is if, and to what extent, myofibers in human’s exhibit signs of necrosis after contraction- or exercise-induced muscle injury. Suffice to say, this issue is controversial and not easily resolved because of inherent limitations of conducting human research. Namely, the tissue obtained through a muscle biopsy represents a very small fraction of the affected whole muscle, which contributes to the large variability in dependent measures that is often observed in human studies [217]. 

Many studies have examined histological and ultrastructural changes in myofibers after voluntary continuous exercise (e.g., downhill running and eccentric cycling) or repetitive lengthening contractions of an isolated muscle (e.g., bicep brachii and quadriceps). In some of these studies, myofiber necrosis was indicated by extensive, localized disruption to sarcomeres within multiple myofibrils [7,11,12,13] and the presence of cells within a few myofibers [218,219,220]. Extensive injury resulting from voluntary lengthening contractions has also been reported to be accompanied by the central position of myonuclei in a localized region of some myofibers during recovery [11,13,220]. However, as detailed in a narrative review [219], the majority of human studies report that exercise-induced muscle injury is not associated with overt signs of myofiber injury/necrosis in transverse sections.

Indices of myofiber injury/necrosis and muscle regeneration in humans have also been assessed before and after electrically stimulated lengthening contractions [221,222,223]. Using this model, Mackey et al. [222] reported that the percentage of necrotic myofibers in transverse sections peaked at 7 days of recovery (median = 4.5%; range = 0.0–15.8%). At 30 days of recovery, the percentage of regenerating myofibers (median = 2.8%; range = 0.0–12.4%) as well as the percentage of myofibers expressing neonatal (median = 16.5%, range = 0.2–75.3%) or embryonic (median = 2.3%, range = 0.0–44.0%) myosin in transverse sections were elevated above control levels [222]. Mackey et al. [223] also analyzed portions of single myofibers isolated from biopsied muscle at 7 and 30 days of recovery from electrically stimulated lengthening contractions. They reported zones of myofiber necrosis and regeneration, as well as the presence of satellite cells and myotubes within a basal lamina at 7 days of recovery [223]. Taken together, the findings of Mackey et al. [222,223] indicate that myotubes/regenerating myofibers form after electrically stimulated lengthening contractions in a manner that reflects the magnitude of myofiber necrosis. 

### 6.3. Summary

Regenerating myofibers can appear in skeletal muscles of animals and humans after contraction- or exercise-induced muscle injury. Their appearance, however, is limited to experimental models that cause overt signs of myofiber injury/necrosis. The magnitude of regenerating myofiber formation after contraction- or exercise-induced muscle injury appears to be proportional to the number of myofibers that displayed moderate-to-severe signs of injury/necrosis. The time scale of regenerating myofiber formation and maturation after injurious contractions, however, remains unresolved. In particular, the limited evidence is inconclusive on the premise that central myonuclei relocate to a peripheral position after injurious contractions. 

The extent to which regenerating myofibers after injurious contractions reflect muscle regeneration in response to complete or segmental necrosis of myofibers also remains unclear. Insight into this issue could be gained by isolating single myofibers after injurious contractions and examining them for indices of myonuclear positioning. Contraction-induced segmental necrosis would be predicted to yield myofibers that have one or more segments containing nuclear chains. The number of segments with nuclear chains and the length of nuclear chains within a segment could be used as a readout of the location and size of the focal lesion/repair of the affected myofiber. On the other hand, nuclear chains would be expected to be found throughout the length of myofibers after injurious contractions, if myofibers underwent complete necrosis prior to regenerating myofiber formation. 

## 7. Concluding Perspective

Myofiber formation and maturation are critically important in establishing muscle structure during embryonic and postnatal myogenesis. Similarly, regenerating myofiber formation and maturation is essential to restoring muscle structure when myofibers undergo necrosis after trauma or injurious contractions. Importantly, the phenotype of myofibers maturing in the embryo, fetus, and during post-natal development are distinctly different from the phenotype of regenerating myofibers during the course of regeneration. Suffice to say, the position, organization, and number of myonuclei, as well as myofiber hypertrophy and morphology during muscle development and regeneration are not identical. Such differences indicate that there are cellular and molecular processes of myofiber maturation that are unique to muscle regeneration. Stated differently, although myofiber maturation during muscle development and regeneration share common processes, such as myogenic cell proliferation, myonuclear accretion, and myofiber hypertrophy, the actual molecules and cell types that regulate such processes are likely to be different between muscle development and regeneration. 

Differences in myofiber maturation during muscle development and regeneration, if acknowledged, are typically explained by differences in the soluble and cellular environment in which myofibers are maturing. Superficially, this is a satisfactory explanation as probing deeper puts one at the edge of the abyss. That is, a large number of genes, proteins, and cell types could be differentially operating when myofibers are maturing during muscle development and regeneration. Although recent work using advanced techniques in transcriptomics have begun to decipher the complex milieu of regenerating muscle [143,224,225,226], they have also most exclusively focused of the initial stages of muscle regeneration. While waiting for evidence of the cellular and molecular processes that govern regenerating myofiber maturation, one should not assume that the processes of regenerating myofiber maturation are analogous to those that mediate myotube/myofiber maturation during in vitro, embryonic, and postnatal myogenesis. Doing so will result in a misguided understanding of regenerating myofiber maturation and impede progress towards the discovery of processes that are unique to muscle regeneration. 

A similar cautionary approach was used in the context of embryonic progenitor cells and adult satellite cells. This proved to be beneficial to advancing knowledge as it resulted in the discovery of the molecular signature of satellite cells, the molecular regulation of myoblasts differentiation, as well as the identification of multiple cell types, cytokines, and signaling pathways that regulate the early stages of muscle regeneration. In fact, it seems that research targeting the early stages of muscle regeneration has resulted in a knowledge base that exceeds current knowledge of the cell types, cytokines, and cell signaling pathways that regulate embryonic and postnatal myogenesis. The hope is that future research targeting the later stages of muscle regeneration will lead to the discovery of cellular and molecular processes of regenerating myofiber maturation. Such discoveries could lead to the development of therapeutics for promoting regeneration when aging, trauma, or disease alters the soluble and cellular environment of skeletal muscle and/or the intrinsic properties of myogenic cells.

## Figures and Tables

**Figure 1 ijms-24-12545-f001:**
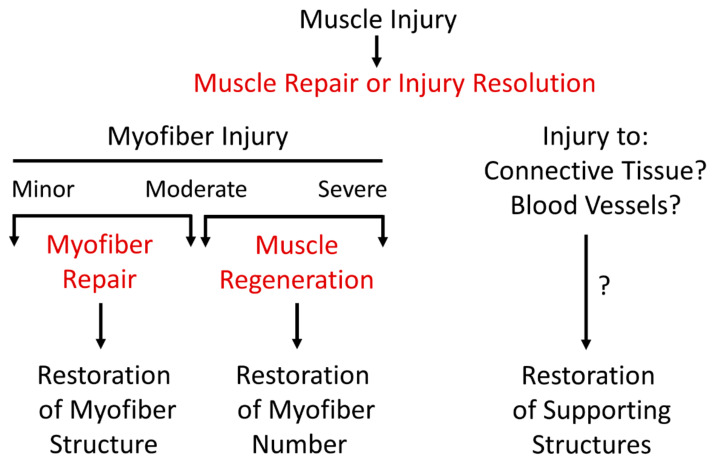
Illustration of nomenclature. Muscle repair and injury resolution are terms that broadly describe processes that restore structure and function to injured muscle. Structural abnormalities within myofibers after muscle injury can range from minor to severe and other components of skeletal muscle could show signs of injury. Minor to moderate injury to myofibers is thought to be reversible; whereas more severe injury results in myofiber necrosis. Myofiber repair defines processes that restore structure to injured myofibers and hence prevents their necrosis. When injured beyond repair, myofibers undergo complete necrosis, and processes of muscle regeneration are initiated. Muscle regeneration restores myofiber number by creating new, replacement myofibers called regenerating myofibers and by promoting their maturation.

**Figure 2 ijms-24-12545-f002:**
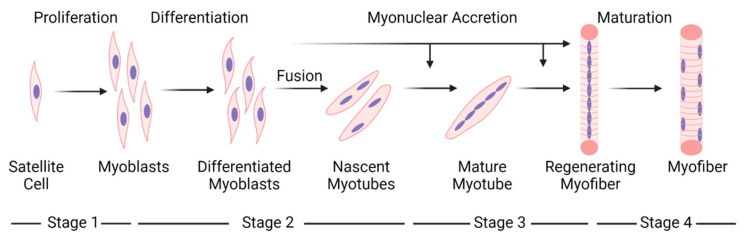
Paradigm for muscle regeneration in response to complete necrosis of myofibers. Muscle regeneration commences with the proliferation of satellites cells, which gives rise to myoblasts that also proliferate (stage 1). Myoblasts differentiate and then adhere and fuse to each other to produce nascent myotubes (stage 2). Myotubes add myonuclei (myonuclei accretion) via myoblast or myotube fusion and ultimately become regenerating myofibers (stage 3). Regenerating myofibers are defined by the central position of myonuclei. Regenerating myofibers mature and become normal sized myofibers with peripheral myonuclei (stage 4).

**Figure 3 ijms-24-12545-f003:**
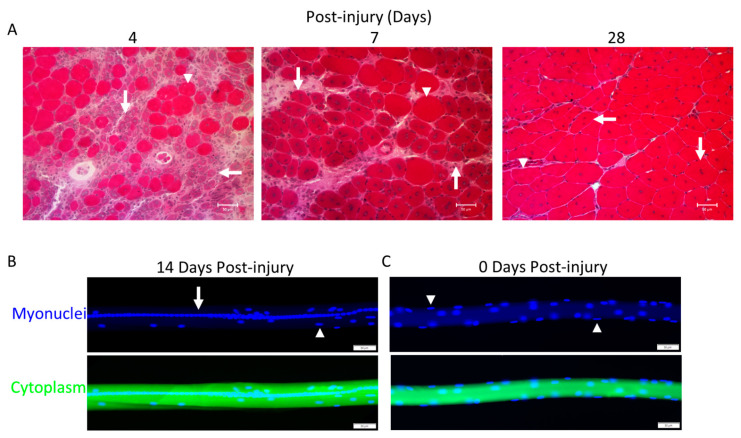
Position of myonuclei in regenerating and normal myofibers. (**A**) Hematoxylin and eosin stained transverse sections of mouse tibialis anterior muscles at 4, 7, and 28 days after chemical (barium chloride) trauma. Regenerating myofibers are indicated by the central position of myonuclei (arrow). Non-regenerating myofibers (arrowhead) after chemical trauma are indicated by the absence of central myonuclei. Calibration bar = 50 µm. (**B**) Z-plane image of myonuclei (blue) in a regenerating myofiber collected from a mouse tibialis anterior muscle 14 days after chemical (barium chloride) trauma. The cytoplasm of the myofiber was delineated via formaldehyde-induced fluorescence (green). Central myonuclei (arrow) are aligned in a linear array (nuclear chain). Some of the myonuclei in the regenerating myofiber are in a peripheral location (arrowhead). Scale bar = 50 µm. (**C**) Z-plane image of myonuclei (blue) in a control (non-regenerating) myofiber (0 days post-injury). Nearly all of the myonuclei in control myofibers are normally in peripheral position (arrowhead). The cytoplasm is indicated by formaldehyde-induced fluorescence (green). Scale bar = 50 µm. All images were obtained using procedures described in Martin et al. (Ref. [86]).

**Figure 4 ijms-24-12545-f004:**
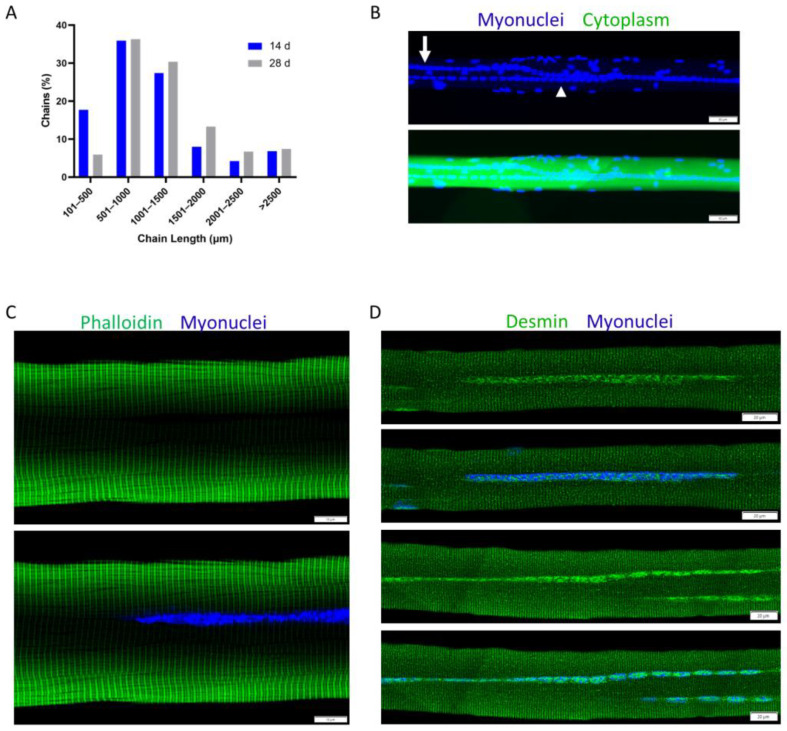
Nuclear chains in regenerating myofibers. (**A**) Frequency distribution plot of individual nuclear chain length (µm) in regenerating myofibers obtained from mouse tibialis anterior muscles at 14 (*n* = 229 nuclear chains in 45 regenerating myofibers) and 28 (*n* = 132 nuclear chains in 35 regenerating myofibers) days after chemical (barium chloride) trauma. Nuclear chain length was measured and reported in Martin et al. (Ref. [86]). (**B**) Image (z-plane) of a regenerating myofiber containing parallel rows of nuclear chains (arrow) and a bifurcated nuclear chain (arrowhead). The cytoplasm is indicated by formaldehyde-induced fluorescence (green). Scale bar = 50 µm. The methodology used to obtain the image is described in Martin et al. (Ref. [86]). (**C**) Image (z-plane) of a regenerating myofiber stained with phalloidin to delineate F-actin (green) within myofibrils. Myonuclei (blue) are positioned between myofibrils (arrow). Scale bar = 10 µm. The methodology used to obtain the image is described in Buckley et al. (Ref. [90]). (**D**) Images (z-plane) of desmin (green) and myonuclei (blue) in regenerating myofibers. Scale bar = 20 µm. Images presented were obtained using procedures described in Martin et al. (Ref. [86]).

**Figure 5 ijms-24-12545-f005:**
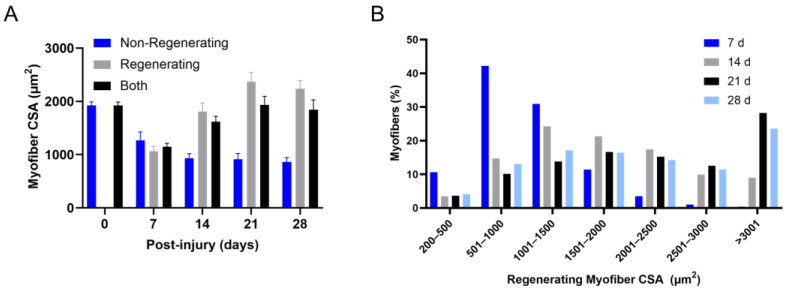
Radial size of myofibers after chemical (barium chloride) trauma. Tibialis anterior muscles of mice were injected with barium chloride, and the cross-sectional area (CSA) of all myofibers was measured and reported in Martin et al. (Ref. [86]). (**A**) The average (±SEM) CSA for regenerating myofiber, non-regenerating, and all myofibers (both) at 7, 14, 21, and 28 d post-injury (*n* = 8 muscles/group). (**B**) Frequency distribution plot of regenerating myofiber CSA at 7, 14, 21, and 28 days post-injury. Data are plotted as a percentage of the total number of regenerating myofibers at each post-injury time point. The total number of regenerating myofibers analyzed was 12,114.

**Figure 6 ijms-24-12545-f006:**
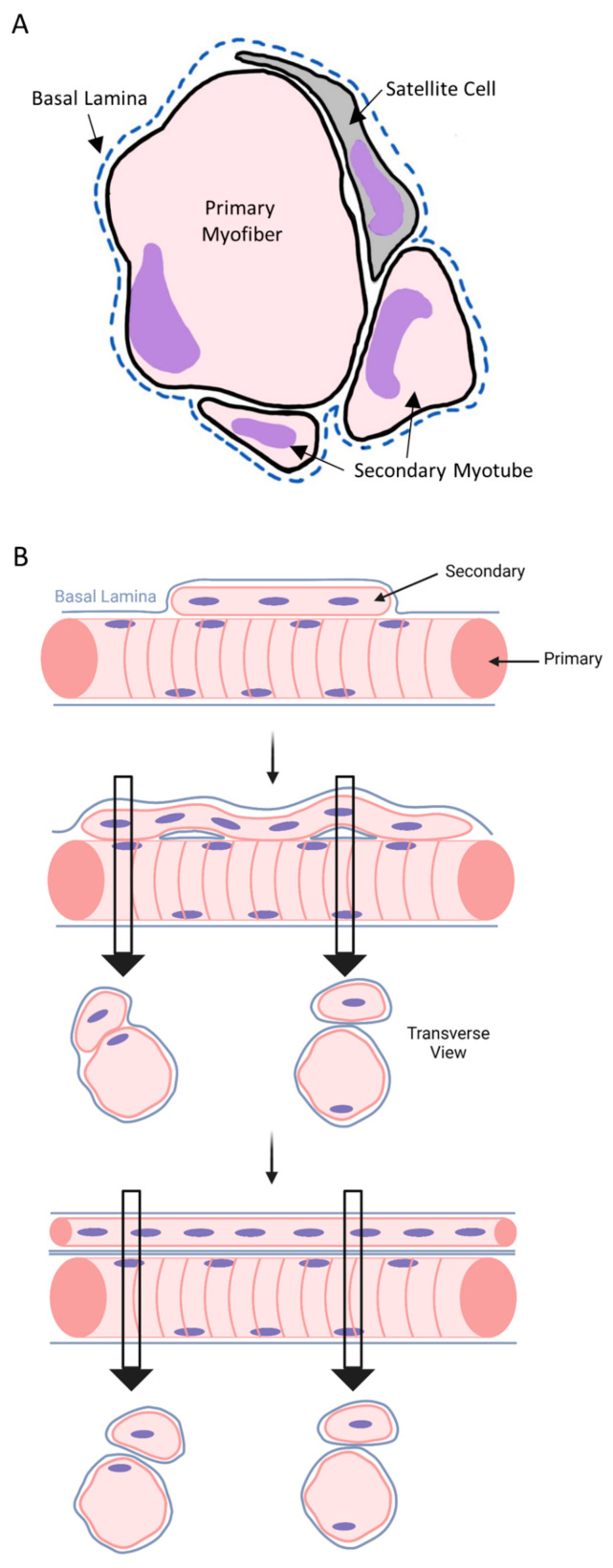
Illustrations of primary and secondary myotubes during embryonic myogenesis. (**A**) Transverse view of primary and secondary myotubes/myofiber, as well as a satellite cell enclosed in a single basal lamina. Adapted from Ontell et al. (Ref. [162]). (**B**) Longitudinal and transverse view of primary and secondary myotubes/myofibers that are in the process of separating from each other during the course of embryonic myogenesis. Arrow indicates the region of the primary and secondary myotube/myofiber depicted in the transverse view. Adapted from Harris (Ref. [160]).

**Figure 7 ijms-24-12545-f007:**
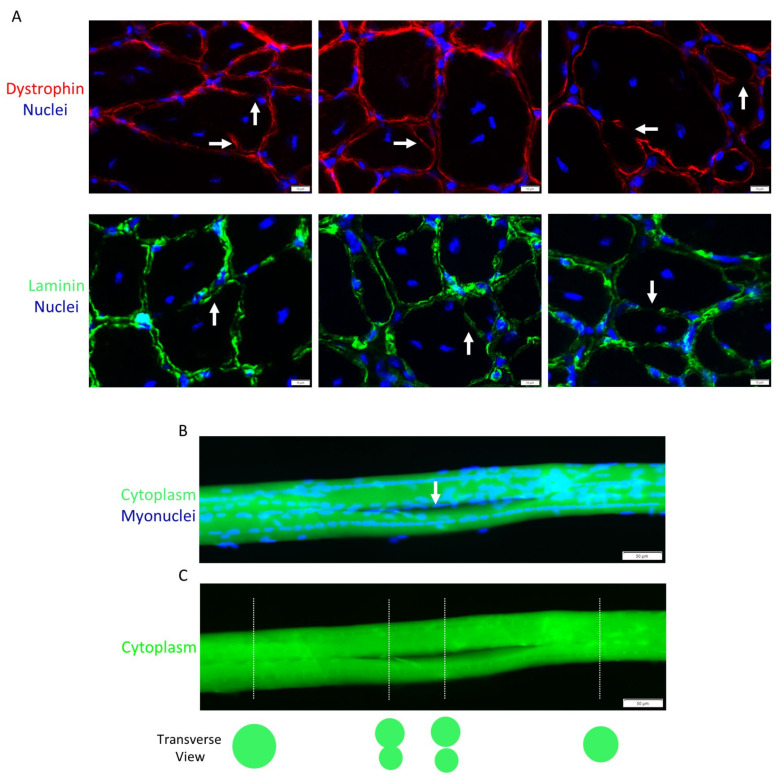
‘Split’ and branched myofibers in tibialis anterior muscles of mice after chemical (barium chloride) trauma in mice. (**A**) Images of dystrophin (red), laminin (green), and nuclei (blue) at 14 or 28 days of recovery. Arrow indicates myofibers with a ‘split’ morphology. Calibration bar = 10 µm. (**B**) Image (z-plane) of myonuclei (blue) in a branched myofiber with a ‘split’/fissure (arrow). The cytoplasm is indicated by formaldehyde-induced fluorescence (green). Calibration bar = 50 µm. (**C**) Image (z-plane) of a branched myofiber (shown in (**B**)) with dashed lines and a cartoon below them to illustrate the morphology of the branched myofiber (green) in transverse sections. Calibration bar = 50 µm. All images were obtained using procedures described in Martin et al. (Ref. [86]).

**Figure 8 ijms-24-12545-f008:**
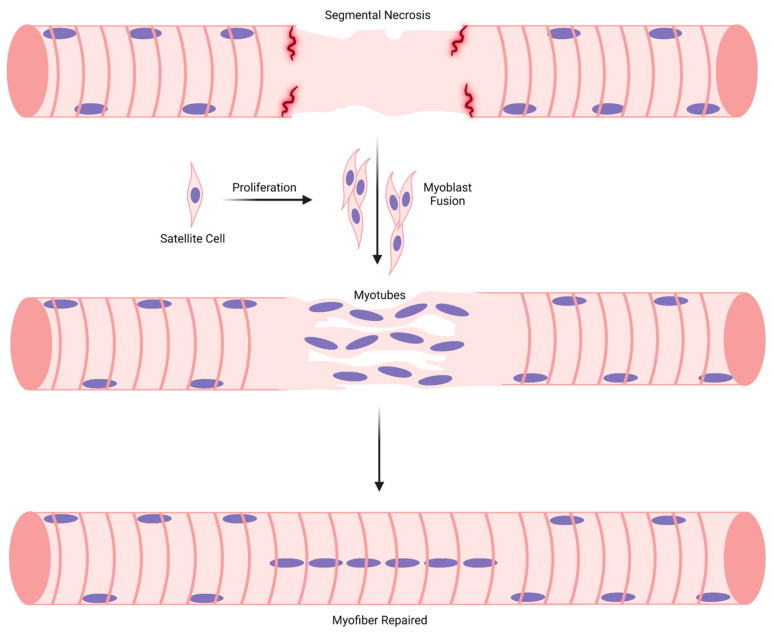
Paradigm for muscle regeneration in response to segmental necrosis of myofibers. Myoblasts, derived from the proliferation of satellite cells, accumulate in the necrotic zone of a myofiber and fuse to form myotubes. Myotube formation and their subsequent maturation in the localized region restores continuity and structure to the affected myofiber, resulting in myofiber repair.

**Figure 9 ijms-24-12545-f009:**
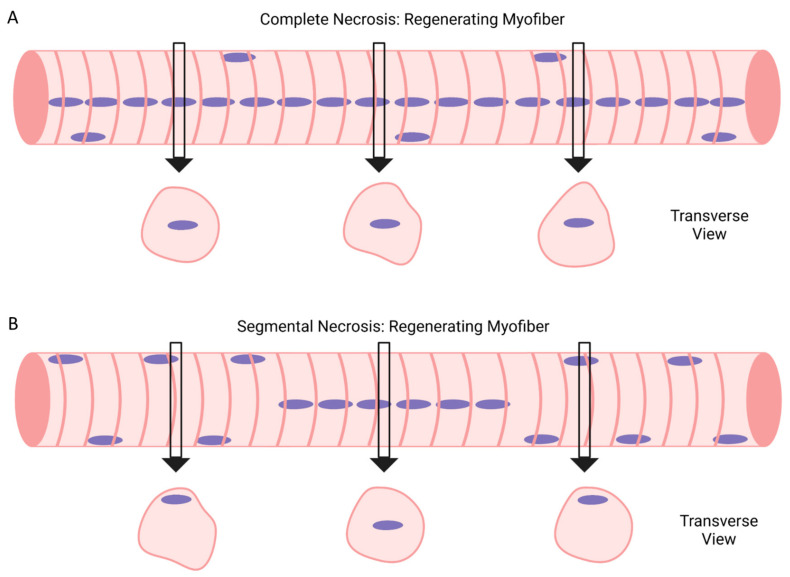
Illustration of regenerating myofibers formed after different types of myofiber necrosis. (**A**) Complete myofiber necrosis. (**B**) Segmental myofiber necrosis. The location of central myonuclei/nuclear chains along the length of a myofiber dictates the extent to which regenerating myofibers are detectable in transverse planes of skeletal muscle.

**Table 1 ijms-24-12545-t001:** Characteristics of Myotubes and Regenerating Myofibers.

	Myotube	Regenerating Myofiber
Central Myonuclei	Yes	Yes
Number of Myonuclei	2–20	Hundreds
Number of Myofibrils	Few	Many
Radial Size	Small	Small to Large
Embryonic Myosin	Yes	Yes, transient

**Table 2 ijms-24-12545-t002:** Regenerating Myofibers after Lengthening Contractions in Rodents.

1st Author	Ref #	Muscle	Species	# Contractions	Recovery Time Point (Weeks)					
					1	2	3	4	6	8
Keller	[197]	SOL	Mice	450	↑ #					
Koh	[208]	EDL	Mice	75		15%			8%	
Lovering	[194]	TA	Rat	150	5%	40%	10%			
Lovering	[193]	TA	Rat	15		40%				
Paul	[179]	EDL	Mice	175–225				30%		
Pizza	[192]	EDL	Mice	75	5%	18%				
Rader	[212]	GAST	Mice	225				60%		60%
Rathbone	[210]	TA	Mice	150						↑ #

SOL = soleus, EDL = extensor digitorum longus, TA = tibialis anterior, GAST = gastrocnemius; ↑ # = Absolute number of regenerating myofibers elevated above control levels.

## Data Availability

Data sets presented are available upon request to the corresponding author.

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
