# Peer review of "Regenerating Myofibers after an Acute Muscle Injury: What Do We Really Know about Them?"

_ijms, 2023, doi:10.3390/ijms241612545_

Round 1

Reviewer 1 Report

The review by Pizza and Buckley focuses on regenerating myofibers after acute muscle injury as compared to maturation of myofibers during muscle development. They highlight the phenotype of regenerating myofibers and provide an overview of the paradigm of muscle regeneration. The regulation of the early stages of muscle regeneration by molecular and cell signaling pathways is expertly summarized in a number of published reviews elsewhere, but references are not provided here, and discussion of specific molecules that regulate aspects of muscle regeneration is intentionally minimal. They compare myonuclear positioning and accretion and myofiber hypertrophy during muscle development and regeneration, providing a framework for identifying cellular and molecular processes of myofiber maturation during muscle regeneration.
Thus, references to many of the published review articles on molecular and cell signaling pathways in muscle regeneration are not provided and therefore not helpful to the reader. At the very least, the sentence on page 2, lines 26-27, should include the expert reviews published in the past five years; given the aims and scope of IJMS for molecular research in molecular biology, reference should also be made to recent reviews on the molecular mechanisms of muscle regeneration.

Regarding Figures 3, 4, 5, and 7, are these examples of the authors' own experiments? If so, details of the experiments, including proper controls and ethical considerations, should be provided. If not, the source of the original figure should be indicated.

Reviewer 2 Report

The authors highlighted very well the aspects related to muscle regeneration after an acute muscle injury and brought arguments that differentiate the formation of myofibers in this case compared to the myogenesis that takes place during embryonic/postnatal development. It is obvious that the authors have experience in studies related to muscle regeneration, given their works and the results cited in this review. The observations and questions regarding many aspects of muscle regeneration are pertinent, welcome and encourage further studies in this field.

However, I have some observations.

Chapter 3. Figure 3 does not cite the source. Even if the figures are original and unpublished, this should be specified.

Chapter 4. Chapter 4 has a title that includes everything described in chapters 5-11. This chapter should be an introduction to the following chapters. Thus, chapters 5-11 should be subchapters of chapter 4 (for example, chapter 5 should be subchapter 4.1, 6, become 4.2 and so on).

Chapter 5. Page 9, row 7. A chapter 6 appears: Myonuclear positioning during trauma-induced muscle regeneration - with non-bold letters. Of course, the chapter numbers should be revised. Probably because of this mistake, the references to Figure 2 in the text are actually references to Figure 3!

Figure 4B does not cite the source.

The explanation from page 11, line 7, is expressed incorrectly: "treated with histological stains". It would be better: "transverse sections stained with H&E (Fig 3A), or immunolabeled to highlight cytoskeletal proteins such F-actin or an isoform of myosin".

Chapter 6. Page 12, row 9-10. What does Px mean? This meaning is explained in reference 110 (P=postnatal mouse age in days), it should also be specified here in the text.

Very good observations regarding myonuclear accretion in the postnatal physiological events.

Chapter 8. Interesting observations. They can have an impact from a medical point of view, taking into account the differences that occur in the regeneration of different types of muscles after trauma.

Why are the figures cited in the text sometimes bold, sometimes not?

Chapter 11. Page 23, row 29 – “When considering myofiber formation during embryonic myogenesis…..”: It is not clear what the authors are referring to, the difference between split/branched myofibers in regeneration after trauma, compared to what is observed in embryonic myogenesis, or is it a mistake. The comparison term is missing. It should be reworded.

Chapter 12. Figure 9. The letters A and B in the figure are missing.
